# Latent Weights Do Not Exist: Rethinking Binarized Neural Network Optimization

**Koen Helwegen**[1]**, James Widdicombe**[1]**, Lukas Geiger**[1]**, Zechun Liu**[2]**, Kwang-Ting Cheng**[2]**, and Roeland Nusselder**[1]

[1]Plumerai Research
{koen, james, lukas, roeland}@plumerai.com
[2]Hong Kong University of Science and Technology
zliubq@connect.ust.hk, timcheng@ust.hk

## Abstract

Optimization of Binarized Neural Networks (BNNs) currently relies on real-valued latent weights to accumulate small update steps. In this paper, we argue that these latent weights cannot be treated analogously to weights in real-valued networks. Instead their main role is to provide inertia during training. We interpret current methods in terms of inertia and provide novel insights into the optimization of BNNs. We subsequently introduce the first optimizer specifically designed for BNNs, Binary Optimizer (Bop), and demonstrate its performance on CIFAR-10 and ImageNet. Together, the redefinition of latent weights as inertia and the introduction of Bop enable a better understanding of BNN optimization and open up the way for further improvements in training methodologies for BNNs. Code is available at: `https://github.com/plumerai/rethinking-bnn-optimization`.

## 1 Introduction

Society can be transformed by utilizing the power of deep learning outside of data centers: self-driving cars, mobile-based neural networks, smart edge devices, and autonomous drones all have the potential to revolutionize everyday lives. However, existing neural networks have an energy budget which is far beyond the scope for many of these applications. Binarized Neural Networks (BNNs) have emerged as a promising solution to this problem. In these networks both weights and activations are restricted to $\{-1, +1\}$, resulting in models which are dramatically less computationally expensive, have a far lower memory footprint, and when executed on specialized hardware yield in a stunning reduction in energy consumption. After the pioneering work on BinaryNet [1] demonstrated such networks could be trained on a large task like ImageNet [2], numerous papers have explored new architectures [3–7], improved training methods [8] and sought to develop a better understanding of their properties [9].

The understanding of BNNs, in particular their training algorithms, have been strongly influenced by knowledge of real-valued networks. Critically, all existing methods use "latent" real-valued weights during training in order to apply traditional optimization techniques. However, many insights and intuitions inspired by real-valued networks do not directly translate to BNNs. Overemphasizing the connection between BNNs and their real-valued counterparts may result in cumbersome methodologies that obscure the training process and hinder a deeper understanding.

In this paper we develop an alternative interpretation of existing training algorithms for BNNs, and subsequently, argue that latent weights are not necessary for gradient-based optimization of BNNs. We introduce a new optimizer based on these insights, which, to the best of our knowledge is the first optimizer designed specifically for BNNs, and empirically demonstrate its performance on CIFAR-10

[10] and ImageNet. Although we study the case where both activations and weights are binarized, the ideas and techniques developed here concern only the binary weights and make no assumptions about the activations, and hence can be applied to networks with activations of arbitrary precision.

The paper is organized as follows. In Section 2 we review existing training methods for BNNs. In Section 3 we give a novel explanation of why these techniques work as well as they do and suggest an alternative approach in Section 4. In Section 5 we give empirical results of our new optimizer on CIFAR-10 and ImageNet. We end by discussing promising directions in which BNN optimization may be further improved in Section 6.

## 2    Background: Training BNNs with Latent Weights

Consider a neural network, $y = f(x, w)$, with weights, $w \in \mathbb{R}^n$, and a loss function, $L(y, y_{\text{label}})$, where $y_{\text{label}}$ is the correct prediction corresponding to sample $x$. We are interested in finding a binary weight vector, $w_{\text{bin}}^{\star}$, that minimizes the expected loss:

$$w_{\text{bin}}^{\star} = \text{argmin}_{w_{\text{bin}} \in \{-1, +1\}^n} \mathbb{E}_{x,y} \left[ L \left( f(x, w_{\text{bin}}), y_{\text{label}} \right) \right]. \tag{1}$$

In contrast to traditional, real-valued supervised learning, Equation 1 adds the additional constraint for the solution to be a binary vector. Usually, a global optimum cannot be found. In real-valued networks an approximate solution via Stochastic Gradient-Descent (SGD) based methods are used instead.

This is where training BNNs becomes challenging. Suppose that we can evaluate the gradient $\frac{\partial L}{\partial w}$ for a given tuple $(x, w, y)$. The question then is how can we use this gradient signal to update $w$, if $w$ is restricted to binary values?

Currently, this problem is resolved by introducing an additional real-valued vector $\tilde{w}$ during training. We call these **latent weights**. During the forward pass we binarize the latent weights, $\tilde{w}$, deterministically such that

$$w_{\text{bin}} = \text{sign}(\tilde{w}) \qquad \text{(forward pass)}. \tag{2}$$

The gradient of the sign operation vanishes almost everywhere, so we rely on a "pseudo-gradient" to get a gradient signal on the latent weights, $\tilde{w}$ [1, 11]. In the simplest case this pseudo-gradient, $\Phi$, is obtained by replacing the binarization during the backward pass with the identity:

$$\Phi(L, \tilde{w}) := \frac{\partial L}{\partial w_{\text{bin}}} \approx \frac{\partial L}{\partial \tilde{w}} \qquad \text{(backward pass)}. \tag{3}$$

This simple case is known as the "Straight-Through Estimator" (STE) [12, 11]. The full optimization procedure is outlined in Algorithm 1. The combination of pseudo-gradient and latent weights makes it possible to apply a wide range of known methods to BNNs, including various optimizers (Momentum, Adam, etc.) and regularizers (L2-regularization, weight decay) [13–15].

Latent weights introduce an additional layer to the problem and make it harder to reason about the effects of different optimization techniques in the context of BNNs. A better understanding of latent weights will aid the deployment of existing optimization techniques and can guide the development of novel methods.

For the sake of completeness we should mention there exists a closely related line of research which considers stochastic BNNs [16, 17]. These networks fall outside the scope of the current work and in the remainder of this paper we focus exclusively on fully deterministic BNNs.

## 3    The Role of Latent Weights

Latent weights absorb network updates. However, due to the binarization function modifications do not alter the behavior of the network unless a sign change occurs. Due to this, we suggest that the latent weight can be better understood when thinking of its sign and magnitude separately:

$$\tilde{w} = \text{sign}(\tilde{w}) \cdot |\tilde{w}| =: w_{\text{bin}} \cdot m, \quad w_{\text{bin}} \in \{-1, +1\}, m \in [0, \infty). \tag{4}$$

The role of the magnitude of the latent weights, $m$, is to provide *inertia* to the network. As the inertia grows, a stronger gradient-signal is required to make the corresponding binary weight flip. Each

**Algorithm 1:** Training procedure for BNNs using latent weights. Note that the optimizer $\mathcal{A}$ may be stateful, although we have suppressed the state in our notation for simplicity.

---

**input :** Loss function $L(f(x; w), y)$, Batch size $K$
**input :** Optimizer $\mathcal{A} : g \mapsto \delta_w$, learning rate $\alpha$
**input :** Pseudo-Gradient $\Phi : L(f(x; w), y) \mapsto g \in \mathbb{R}^n$

initialize $\tilde{w} \leftarrow \tilde{w}^0 \in \mathbb{R}^n$;

**while** stopping criterion not met **do**

    Sample minibatch $\{x^{(1)}, ..., x^{(K)}\}$ with labels $y^{(k)}$;
    Perform forward pass using $w_{\text{bin}} = \text{sign}(\tilde{w})$;
    Compute gradient: $g \leftarrow \frac{1}{K} \Phi \sum_k L(f(x^{(k)}; w_{\text{bin}}), y^{(k)})$;
    Update latent weights $\tilde{w} \leftarrow \tilde{w} + \alpha \cdot \mathcal{A}(g)$;

**end**

---

binary weight, $w_{\text{bin}}$, can build up inertia, $m$, over time as the magnitude of the corresponding latent weight increases. Therefore, latent weights are not weights at all: they encode both the binary weight, $w_{\text{bin}}$, and a corresponding inertia, $m$, which is really an optimizer variable much like momentum.

We contrast this inertia-based view with the common perception in the literature, which is to see the binary weights as an *approximation to the real-valued weight vector*. In the original BinaryConnect paper, the authors describe the binary weight vector as a discretized version of the latent weight, and draw an analogy to Dropout in order to explain why this may work [18, 19]. Anderson and Berg argue that binarization works because the angle between the binarized vector and the weight vector is small [9]. Li et al. prove that, for a quadratic loss function, in BinaryConnect the *real-valued weights* converge to the global minimum, and argue this explains why the method outperforms Stochastic Rounding [20]. Merolla et al. challenge the view of approximation by demonstrating that many projections, onto the binary space and other spaces, achieve good results [21].

A simple experiment suggests the approximation viewpoint is problematic. After training the BNN, we can evaluate the network using the real-valued latent weights instead of the binarized weights, while keeping the binarization of the activations. If the approximation view is correct, using the real-valued weights should result in a higher accuracy than using the binary weights. We find this is not the case. Instead, we consistently see a comparable or lower train and validation accuracy when using the latent weights, even after retraining the batch statistics.

The concept of inertia enables us to better understand what happens during the optimization of BNNs. Below we review some key aspects of the optimization procedure from the perspective of inertia.

First and foremost, we see that in the context of BNNs, the optimizer is mostly changing the inertia of the network rather than the binary weights themselves. The inertia variables have a stabilizing effect: after being pushed in one direction for some time, a stronger signal in the reverse direction is required to make the weight flip. Meanwhile, clipping of latent weights, as is common practice in the literature, influences training by ceiling the inertia that can be accumulated.

In the optimization procedure defined by Algorithm 1, scaling of the learning rate does not have the role one may expect, as is made clear by the following theorem:

**Theorem 1.** *The binary weight vector generated by Algorithm 1 is invariant under scaling of the learning rate, $\alpha$, provided the initial conditions are scaled accordingly and the pseudo-gradient, $\Phi$, does not depend on $|\tilde{w}|$.*

The proof for Theorem 1 is presented in the appendix. An immediate corollary is that in this setting we can set an arbitrary learning rate for every individual weight as long as we scale the initialization accordingly.

We should emphasize the conditions to Theorem 1 are rarely met: usually latent weights are clipped, and many pseudo-gradients depend on the magnitude of the latent weight. Nevertheless, in experiments we have observed that the advantages of various learning rates can also be achieved by scaling the initialization. For example, when using SGD and Glorot initialization [11] a learning rate of 1 performs much better than $0.01$; but when we multiply the initialized weights by $0.01$ before starting training, we obtain the same improvement in performance.

Theorem 1 also helps to understand why reducing the learning rate after training for some time helps: it effectively increases the already accumulated inertia, thus reducing noise during training. Other techniques that modify the magnitude of update-steps, such as the normalizing aspect of Adam and the layerwise scaling of learning rates introduced in [1], should be understood in similar terms. Note that the ceiling on inertia introduced by weight clipping may also play a role, and a full explanation requires further analysis.

Clearly, the benefits of using Momentum and Adam over vanilla-SGD that have been observed for BNNs [8] cannot be explained in terms of characteristics of the loss landscape (curvature, critical points, etc.) as is common in the real-valued context [22, 14, 23, 24]. We hypothesize that the main effect of using Momentum is to reduce noisy behavior when the latent weight is close to zero. As the latent weight changes signs, the direction of the gradient may reverse. In such a situation, the presence of momentum may avoid a rapid sign change of the binary weight.

## 4    Bop: a Latent-Free Optimizer for BNNs

In this section we introduce the Binary Optimizer, referred to as Bop, which is to the best of our knowledge, the first optimizer designed specifically for BNNs. It is based on three key ideas.

First, the optimizer has only a single action available: flipping weights. Any concept used in the algorithm (latent weights, learning rates, update steps, momentum, etc) only matters in so far as it affects weight flips. In the end, any gradient-based optimization procedure boils down to a single question: how do we decide whether to flip a weight or not, based on a sequence of gradients? A good BNN optimizer provides a concise answer to this question and all concepts it introduces should have a clear relation to weight flips.

Second, it is necessary to take into account past gradient information when determining weight flips: it matters that a signal is *consistent*. We define a gradient signal as the average gradient over a number of training steps. We say a signal is more consistent if it is present in longer time windows. The optimizer must pay attention to consistency explicitly because the weights are binary. There is no accumulation of update steps.

Third, in addition to consistency, there is meaningful information in the *strength* of the gradient signal. Here we define strength as the absolute value of the gradient signal. As compared to real-valued networks, in BNNs there is only a weak relation between the gradient signal and the change in loss that results from a flip, which makes the optimization process more noisy. By filtering out weak signals, especially during the first phases of training, we can reduce this noisiness.

In Bop, which is described in full in Algorithm 2, we implement these ideas as follows. We select consistent signals by looking at an exponential moving average of gradients:

$$m_t = (1 - \gamma)m_{t-1} + \gamma g_t = \gamma \sum_{r=0}^{t} (1 - \gamma)^{t-r} g_r, \qquad (5)$$

where $g_t$ is the gradient at time $t$, $m_t$ is the exponential moving average and $\gamma$ is the *adaptivity rate*. A high $\gamma$ leads to quick adaptation of the exponential moving average to changes in the distribution of the gradient.

It is easy to see that if the gradient $g_t^i$ for some weight $i$ is sampled from a stable distribution, $m_t^i$ converges to the expectation of that distribution. By using this parametrization, $\gamma$ becomes to an extend analogous to the learning rate: reducing $\gamma$ increases the consistency that is required for a signal to lead to a weight flip.

We compare the exponential moving average with a threshold $\tau$ to determine whether to flip each weight:

$$w_t^i = \begin{cases} -w_{t-1}^i & \text{if } |m_t^i| \geq \tau \text{ and } \text{sign}(m_t^i) = \text{sign}(w_{t-1}^i), \\ w_{t-1}^i & \text{otherwise.} \end{cases} \qquad (6)$$

This allows us to control the strength of selected signals in an effective manner. The use of a threshold has no analogue in existing methods. However, similar to using Momentum or Adam to update latent weights, a non-zero threshold avoids rapid back-and-forth of weights when the gradient reverses on a weight flip. Observe that a high $\tau$ can result in weights never flipping despite a consistent gradient pressure to do so, if that signal is too weak.

---
**Algorithm 2:** Bop, an optimizer for BNNs.
---
**input :** Loss function $L(f(x; w), y)$, Batch size $K$
**input :** Threshold $\tau$, adaptivity rate $\gamma$
initialize $w \leftarrow w_0 \in \{-1, 1\}^n$, $m \leftarrow m_0 \in \mathbb{R}^n$ ;
**while** stopping criterion not met **do**
    Sample minibatch $\{x^{(1)}, ..., x^{(K)}\}$ with labels $y^{(k)}$;
    Compute gradient: $g \leftarrow \frac{1}{K} \frac{\partial L}{\partial w} \sum_k L(f(x^{(k)}; w), y^{(k)})$;
    Update momentum: $m \leftarrow (1 - \gamma)m + \gamma g$;
    **for** $i \leftarrow 1$ **to** $n$ **do**
        **if** $|m_i| > \tau$ and $\text{sign}(m_i) = \text{sign}(w_i)$ **then**
            $w_i \leftarrow -w_i$;
        **end**
    **end**
**end**
---

Both hyperparameters, the adaptivity rate $\gamma$ and threshold $\tau$, can be understood directly in terms of the consistency and strength of gradient signals that lead to a flip. A higher $\gamma$ results in a more adaptive moving average: if a new gradient signal pressures a weight to flip, it will require less time steps to do so, leading to faster but more noisy learning. A higher $\tau$ on the other hand makes the optimizer less sensitive: a stronger gradient signal is required to flip a weight, reducing noise at the risk of filtering out valuable smaller signals.

As compared to existing methods, Bop drastically reduces the number of hyperparameters and the two hyperparameters left have a clear relation to weight flips. Currently, one has to decide on an initialization scheme for the latent weights, an optimizer and its hyperparameters, and optionally constraints or regularizations on the latent weights. The relation between many of these choices and weight flipping - the only thing that matters - is not at all obvious. Furthermore, Bop reduces the memory requirements during training: it requires only one real-valued variable per weight, while the latent-variable approach with Momentum and Adam require two and three respectively.

Note that the concept of consistency here is closely related to the concept of inertia introduced in the previous section. If we initialize the latent weights in Algorithm 1 at zero, they contain a sum, weighted by the learning rate, over all gradients. Therefore, its sign is equal to the sign over the weighted average of past gradients. This introduces an undue dependency on old information. Clipping of the latent weights can be seen as an ad-hoc solution to this problem. By using an exponential moving average, we eliminate the need for latent weights, a learning rate and arbitrary clipping; at the same time we gain fine-grained control over the importance assigned to past gradients through $\gamma$.

We believe Bop should be viewed as a basic binary optimizer, similar to SGD in real-valued training. We see many research opportunities both in the direction of hyperparameter schedules and in adaptive variants of Bop. In the next section, we explore some basic properties of the optimizer.

## 5 Empirical Analysis

### 5.1 Hyperparameters

We start by investigating the effect of different choices for $\gamma$ and $\tau$. To better understand the behavior of the optimizer, we monitor the accuracy of the network and the ratio of weights flipped at each step using the following metric:

$$\pi_t = \log \left( \frac{\text{Number of flipped weights at time } t}{\text{Total number of weights}} + e^{-9} \right). \tag{7}$$

Here $e^{-9}$ is added to avoid $\log(0)$ in the case of no weight flips.

The results are shown in Figure 1. We see the expected patterns in noisiness: both a higher $\gamma$ and a lower $\tau$ increase the number of weight flips per time step.

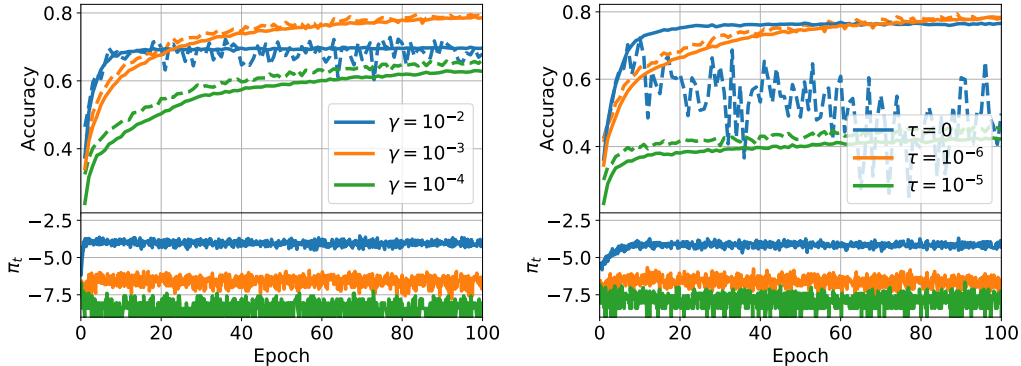

Figure 1: Comparison of training for different values of $\gamma$ and $\tau$ in Bop for BinaryNet on CIFAR-10. The upper panels show accuracy (train: —, validation: – –). The lower panels show $\pi_t$, as defined in Equation (7), for the last layer of the network. On the left side we compare three values for $\gamma$, while keeping $\tau$ fixed at $10^{-6}$. On the right we compare three values for $\tau$, while keeping $\gamma$ fixed at $10^{-3}$. We see that both high $\gamma$ and low $\tau$ lead to rapid initial learning but result in high flip rates, while low $\gamma$ and high $\tau$ result in slow learning and near-zero flip rates.

More interesting is the corresponding pattern in accuracy. For both hyperparameters, we find there is a "sweet spot". Choosing a very low $\gamma$ and high $\tau$ leads to extremely slow learning. On the other hand, overly aggressive hyperparameter settings (high $\gamma$ and low $\tau$) result in rapid initial learning that quickly levels off at a suboptimal training accuracy: it appears the noisiness prevents further learning.

If we look at the validation accuracy for the two aggressive settings $((\gamma, \tau) = (10^{-2}, 10^{-6})$ and $(\gamma, \tau) = (10^{-3}, 0))$, we see the validation accuracy becomes highly volatile in both cases, and deteriorates substantially over time in the case of $\tau = 0$. This suggest that by learning from weak gradient-signals the model becomes more prone to overfit. The observed overfitting cannot simply be explained by a higher sensitivity to gradients from a single example or batch, because then we would expect to observe a similarly poor generalization for high $\gamma$.

These empirical results validate the theoretical considerations that informed the design of the optimizer in the previous section. The behavior of Bop can be easily understood in terms of weight flips. The poor results for high $\gamma$ confirm the need to favor consistent signals, while our results for $\tau = 0$ demonstrate that filtering out weak signals can greatly improve optimization.

## 5.2 CIFAR-10

We use a VGG [25] inspired network architecture, equal to the implementation used by Courbariaux et al. [1]. We scale the RGB images to the interval $[-1, +1]$, and use the following data augmentation during training to improve generalization (as first observed in [26] for CIFAR datasets): 4 pixels are padded on each side, a random $32 \times 32$ crop is applied, followed by a random horizontal flip. During test time the scaled images are used without any augmentation. The experiments were conducted using TensorFlow [27] and NVIDIA Tesla V100 GPUs.

In assessing the new optimizer, we are interested in both the final test accuracy and the number of epochs it requires to achieve this. As discussed in [8], the training time for BNNs is currently far longer than what one would expect for the real-valued case, and is in the order of $500$ epochs, depending on the optimizer. To benchmark Bop we train for $500$ epochs with threshold $\tau = 10^{-8}$, adaptivity rate $\gamma = 10^{-4}$ decayed by $0.1$ every $100$ epochs, batch size $50$, and use Adam with the recommended defaults for $\beta_1$, $\beta_2$, $\epsilon$ [14] and an initial learning rate of $\alpha = 10^{-2}$ to update the real-valued variables in the Batch Normalization layers [28]. We use Adam with latent real-valued weights as a baseline, training for $500$ epochs with Xavier learning rate scaling [18] (as recommended in [8]) using the recommended defaults for $\beta_1$, $\beta_2$ and $\epsilon$, learning rate $10^{-3}$, decayed by $0.1$ every $100$ epochs, and batch size $50$.

The results for the top-1 training and test accuracy are summarized in Figure 2. Compared to the base test accuracy of **90.9**%, Bop reaches **91.3**%. The baseline accuracy was highly tuned using a

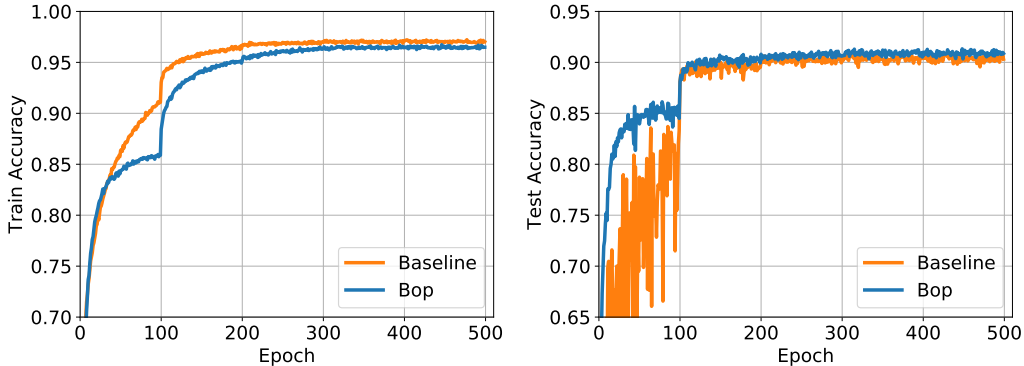

Figure 2: Training and test accuracy history for the models trained. One can see that the results for Bop are competitive with the baseline.

Table 1: Accuracies for Bop on ImageNet for three common BNNs. Results for latent weights are cited from the relevant literature [29, 6, 4].

| Model | Bop (ours) | | Latent weights | |
|---|---|---|---|---|
| | top-1 | top-5 | top-1 | top-5 |
| BinaryNet | **41.1%** | 65.4% | 40.1% | 66.3% |
| XNOR-Net | **45.9%** | 70.0% | 44.2% | 69.2% |
| BiReal-Net | **56.6%** | 79.4% | 56.4% | 79.5% |

extensive random search for the initial learning rate, and the learning rate schedule, and improves the result found in [1] by $1.0\%$.

## 5.3 ImageNet

We test Bop on ImageNet by training three well-known binarized networks from scratch: BinaryNet, a binarized version of Alexnet [29]; XNOR-Net, a improved version of BinaryNet that uses real-valued scaling factors and real-valued first and last layers [6]; and BiReal-Net, which introduced real-valued shortcuts to binarized networks and achieves drastically better accuracy [4].

We train BinaryNet and BiReal-Net for 150 epochs and XNOR-Net for 100 epochs. We use a batch size of $1024$ and standard preprocessing with random flip and resize but no further augmentation. For all three networks we use the same optimizer hyperparameters. We set the threshold to $1 \cdot 10^{-8}$ and decay the adaptivity rate linearly from $1 \cdot 10^{-4}$ to $1 \cdot 10^{-6}$. For the real-valued variables, we use Adam with a linearly decaying learning rate from $2.5 \cdot 10^{-3}$ to $5 \cdot 10^{-6}$ and otherwise default settings ($\beta_1 = 0.9$, $\beta_2 = 0.999$ and $\epsilon = 1 \cdot 10^{-7}$). After observing overfitting for XNOR-Net we introduce a small l2-regularization of $5 \cdot 10^{-7}$ on the (real-valued) first and last layer for this network only. For binarization of the activation we use the STE in BinaryNet and XNOR-Net and the ApproxSign for BiReal-Net, following Liu et al. [4]. Note that as the weights are not binarized in the forward pass, no pseudo-gradient for the backward pass needs to be defined. Moreover, whereas XNOR-net and BiReal-Net effectively binarize to $\{-\alpha, \alpha\}$ by introducing scaling factors, we learn strictly binary weight kernels.

The results are shown in Table 1. We obtain competitive results for all three networks. We emphasize that while each of these papers introduce a variety of tricks, such as layer-wise scaling of learning rates in [1], scaled binarization in [6] and a multi-stage training protocol in [4], we use almost identical optimizer settings for all three networks. Moreover, our improvement on XNOR-Net demonstrates scaling factors are not necessary to train BNNs to high accuracies, which is in line with earlier observations [30].

# 6  Discussion

In this paper we offer a new interpretation of existing deterministic BNN training methods which explains latent real-valued weights as encoding inertia for the binary weights. Using the concept of inertia, we gain a better understanding of the role of the optimizer, various hyperparameters, and regularization. Furthermore, we formulate the key requirements for a gradient-based optimization procedure for BNNs and guided by these requirements we introduce Bop, the first optimizer designed for BNNs. With this new optimizer, we have exceeded the state-of-the-art result for BinaryNet on CIFAR-10 and achieved a competitive result on ImageNet for three well-known binarized networks.

Our interpretation of latent weights as inertia differs from the common view of BNNs, which treats binary weights as an approximation to latent weights. We argue that the real-valued magnitudes of latent weights should not be viewed as weights at all: changing the magnitudes does not alter the behavior of the network in the forward pass. Instead, the optimization procedure has to be understood by considering under what circumstances it flips the binary weights.

The approximation viewpoint has not only shaped understanding of BNNs but has also guided efforts to improve them. Numerous papers aim at reducing the difference between the binarized network and its real-valued counterpart. For example, both the scaling introduced by XNOR-Net (see eq. (2) in [6]) and DoReFa (eq. (7) in [31]), as well as the magnitude-aware binarization introduced in Bi-Real Net (eq. (6) in [4]) aim at bringing the binary vector closer to the latent weight. ABC-Net maintains a single real-valued weight vector that is projected onto multiple binary vectors (eq. (4) in [5]) in order to get a more accurate approximation (eq. (1) in [5]). Although many of these papers achieve impressive results, our work shows that improving the approximation is not the only option. Instead of improving BNNs by reducing the difference with real-valued networks during training, it may be more fruitful to modify the optimization method in order to better suit the BNN.

Bop is the first step in this direction. As we have demonstrated, it is conceptually simpler than current methods and requires less memory during training. Apart from this conceptual simplification, the most novel aspect of Bop is the introduction of a threshold $\tau$. We note that when setting $\tau = 0$, Bop is mathematically similar to the latent weight approach with SGD, where the moving averages $m$ now play the role of latent variables.

The threshold that is used introduces a dependency on the absolute magnitude of the gradients. We hypothesize the threshold helps training by selecting the most important signals and avoiding rapid changes of a single weight. However, a fixed threshold for all layers and weights may not be the optimal choice. The success of Adam for real-valued methods and the invariance of latent-variable methods to the scale of the update step (see Theorem 1) suggest some form of normalization may be useful.

We see at least two possible ways to modify thresholding in Bop. First, one could consider layer-wise normalization of the exponential moving averages. This would allow selection of important signals within each layer, thus avoiding situations in which some layers are noisy and other layers barely train at all. A second possibility is to introduce a second moving average that tracks the magnitude of the gradients, similar to Adam.

Another direction in which Bop may be improved is the exploration of hyperparameter schedules. The adaptivity rate, $\gamma$, may be viewed as analogous to the learning rate in real-valued optimization. Indeed, if we view the moving averages, $m$, as analogous to latent weights, lowering $\gamma$ is analogous to decreasing the learning rate, which by Theorem 1 increases inertia. Reducing $\gamma$ over time therefore seems like a sensible approach. However, any analogy to the real-valued setting is imperfect, and it would be interesting to explore different schedules.

Hyperparameter schedules could also target the threshold, $\tau$, (or an adaptive variation of $\tau$). We hypothesize one should select for strong signals (i.e. high $\tau$) in the first stages of training, and make training more sensitive by lowering $\tau$ over time, perhaps while simultaneously lowering $\gamma$. However, we stress once again that such intuitions may prove unreliable in this unexplored context.

More broadly, the shift in perspective presented here opens up many opportunities to further improve optimization methods for BNNs. We see two areas that are especially promising. The first is regularization. As we have argued, it is not clear that applying L2-regularization or weight decay to the latent weights should lead to any regularization at all. Applying Dropout to BNNs is also problematic. Either the zeros introduced by dropout are projected onto $\{-1, +1\}$, which is likely to

result in a bias, or zeros appear in the convolution, which would violate the basic principle of BNNs. It would be interesting to see custom regularization techniques for BNNs. One very interesting work in this direction is [32].

The second area where we anticipate further improvements in BNN optimization is knowledge distillation [33]. One way people currently translate knowledge from the real-valued network to the BNN is through initialization of the latent weights, which is becoming increasingly sophisticated [4, 8, 34]. Several works have started to apply other techniques of knowledge distillation to low-precision networks [34–36].

The authors are excited to see how to concept of inertia introduced within this paper influence the future development of the field.

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

## A  Proof for Theorem 1

*Proof.* Consider a single weight. Let $\tilde{w}_t$ be the latent weight at time $t$, $g_t$ the pseudo-gradient and $\delta_t$ the update step generated by the optimizer $\mathcal{A}$. Then:

$$\tilde{w}_{t+1} = \tilde{w}_t + \alpha\delta_t.$$

Now take some positive scalar $C$ by which we scale the learning rate. Replace the weight by $\tilde{v}_t = C\tilde{w}_t$. Since $\text{sign}(\tilde{v}_t) = \text{sign}(\tilde{w}_t)$, the binary weight is unaffected. Therefore the forward pass at time $t$ is unchanged and we obtain an identical pseudo-gradient $g_t$ and update step $\delta_t$. We see:

$$\tilde{v}_{t+1} = \tilde{v}_t + C\alpha\delta_t = C \cdot (\tilde{w}_t + \alpha\delta_t) = C\tilde{w}_{t+1}.$$

Thus $\text{sign}(\tilde{v}_{t+1}) = \text{sign}(\tilde{w}_{t+1})$. By induction, this holds for $\forall t' > t$ and we conclude the BNN is unaffected by the change in learning rate. $\qquad\square$

