[Reviews · NeurIPS 2019]

Reviewer 1



Update 2019-08-07 The authors rebuttal shared with us more convincing ImageNet results. They did not address my other concerns as thoroughly. Nonetheless, I improved my score from 6 to 7. ************************** Some general comments: The paper is well written and easy to read. The introduction clearly states the objective and the context. The text and the 2 pseudo-algorithms clearly explain how the method works, and how it differs from previous methods. I did not feel the need to ask the authors to share their code. The CIFAR-10 experiments are solid and involve a rigorous tuning of the baseline hyperparameters. The ImageNet experiments seem preliminary but are rather promising. Other comments: A) Though the proposed method is more straightforward than previous methods, it is unclear whether it is easier to tune. Some of the previous methods only required to tune ADAM’s alpha. The latent weights were initialized and clipped using Glorot-et-al or He-et-al initialization methods. In comparison, the proposed method, BOP, requires tuning 2 hyperparameters, an adaptivity rate and a global threshold. It would be nice if the method did not require to tune a global threshold (but it is no big deal). Besides, it seems a bit awkward that there is a single global threshold for the whole network. Intuitively, I feel like wide layers may require a smaller threshold than thin layers. B) I agree with the authors that it would be nice if their method used the gradient second moment (like ADAM). C) It seems that the method is very dependent on Batch Normalization to maintain unit variance across the units. Maybe scale the binarized weights (after the sign function) so that the method can work without batch normalization? D) The method currently only works with 1-bit weights. It would be nice if it could also work with ternary, 2-bit or any number of bit weights. It would be even nicer if the infinite bit method was equivalent to SGD or ADAM.

Reviewer 2



As opposed to update the real-valued network as usual and quantize the learned one for feedforward, the inertia concept is useful as it can explain the behavior of the gradient descent-based updates. Eventually, it's natural to understand that a large magnitude in the corresponding real-valued network somehow represents the "confidence" about the sign of that parameter, which will be used as a binary variable in the BNN. I agree with this view. What I'm wondering though is the fact that it is already known that the gradient for a particular weight is not defined by its current value, but by the input values and the backpropagation error associated with it (the multiplication of the two). In that regard, it is obvious that some kind of momentum terms will help improve the speed of convergence, as already known by various momentum-based optimization techniques. The proposed algorithm, Bop, to me sounds like one of those variants, while the authors claim that it is the first optimizer specifically for BNN. More specifically, the update rule in eq (5) sounds familiar to me, as it is equivalent to the accumulated gradients as in the original definition of momentum. Slight difference would be that in the proposed algorithm the accumulated gradients replaces the magnitudes while in the original definition of the momentum method it replaces the gradient update. I wonder what's the main difference between the proposed method and a regular momentum-based approach then. Meanwhile, the experimental results show marginal improvement.

Reviewer 3



This paper addresses the optimization for BNN and provides a novel latent-free optimizer for BNN, which challenges the existing way of using later-weights. This is an interesting and original idea. Specifically, one common way to see BNN training is to view the binary weights as an approximation to real-valued weight vector, this paper argues that the latent weights used in the previous methods are in fact not weights. The paper argues this by introducing a concept of inertia. Motivated from this new insight, one novel optimizer called Bop is introduced. Compared with existing latent-based optimizers, Bop requires less memory. One of my previous concern is that relationship and understanding of latent weight methods from the perspective of Bop method. The author's response made a clear explanation by drawing similarities between the momentum and threshold in mitigating noisy flipping behavior. I agree with this viewpoint. The Experimental results on Cifar10 and ImageNet dataset demonstrates that Bop achieves comparative results with baseline optimizers. The original submit shows that Bop slightly worse results than baseline on ImageNet with 50 epochs. However, in the authors' response, they added results for BiReal-Net on ImageNet from scratch using Bop trained for 200 epochs, which achieves competitive results with reported methods. I think this result is promising. Overall, this paper provides a novel idea of training BNN with latent-free weights, which is interesting and insightful and might open a new way train BNN. The experimental results of the proposed Bop optimizer are comparative with existing methods (Experimental details need to be improved). Future directions in improving the proposed Bop are also given. ------------------------------------ Update 2019-08-07 1. The authors' rebuttal adds results for BiReal-Net on ImageNet from scratch using Bop trained for 200 epochs, which achieves competitive (slightly better) results with reported methods. I think this result is promising. 2. In the rebuttal, the authors explains the connections between the proposed Bop and traditional latent-weight methods to further help the understanding of Bop from the perspective of momentum. It is interesting but I think some further understanding is still needed. I keep my previous score, which is still 7.

[Author Response · NeurIPS 2019]

We thank the reviewers for their comments and constructive feedback. We are happy to see all three reviewers appreciate the novel perspective on BNN training we present. We acknowledge the need for better empirical support of our claims and present further ImageNet experiments below. We then address the concerns of each reviewer.

**ImageNet Results.** We train the BiReal-Net architecture on ImageNet from scratch using Bop. We train for 200 epochs with a batch size of 1024 and use standard preprocessing with random flip and resize but no further augmentation. We use Adam ($\beta_1 = 0.9$, $\beta_2 = 0.999$) for the full-precision weights. We linearly decrease three hyperparameters: $\tau$ from $10^{-7}$ to $10^{-8}$, $\gamma$ from $5 \cdot 10^{-4}$ to $5 \cdot 10^{-7}$ and the Adam learning rate from $2.5 \cdot 10^{-3}$ to $5 \cdot 10^{-6}$. After training we recompute the batch norm statistics over one epoch while keeping the weights fixed. We achieve **56.5%** top-1 and **79.5%** top-5 accuracy, which is comparable to the 56.4% top-1 and 77.2% top-5 accuracy originally reported. Note that the original paper relies on a multi-stage pretraining procedure whereas we train the network from scratch.

**Response to Reviewer #2. Tuning of Bop vs. Latent-Weight Methods.** Ease of use is an important consideration when selecting an optimizer so this is a valid concern. Bop is a novel optimizer that is disconnected from the vast body of experience that has developed around SGD-based methods. Inevitably, it will take time to develop intuition for the newly introduced hyperparameters. However, we are optimistic about the prospect of developing these intuitions as the introduced hyperparameters can be directly related to the training behavior of the network, as exemplified by Figure 1 in the paper. Furthermore, in latent-weight methods tuning alpha is only sufficient after numerous other hyperparameters have been fixed, including the initialization and clipping of the latent weights, the pseudo-gradient of the weight-binarization and the betas in Adam. Theorem 1 demonstrates the relations between these can be non-trivial and their effect can be non-intuitive. **The Role of Batch Normalization (BN).** The reviewer is correct in pointing out the BN is important for our method to work. Please note the BN is used in latent methods as well and that in BiReal-Net style architectures the BN is also important for the forward pass. **Adaptivitiy of Bop.** The reviewer shares our enthusiasm for the prospect of adaptive variants of Bop. We remark that in terms of training behavior, increasing the threshold is equivalent to lowering the gradients and so second order moments and adaptive thresholds are strongly related concepts. There are many open questions here, such as whether adaptivity should be implemented globally, per layer or per weight. We think it is valuable to have Bop, the simplest possible implementation of a BNN optimizer, as a reference point and leave the exploration of more sophisticated methods to future work.

**Response to Reviewer #3. Relation to Existing Methods.** The key novelty of Bop is the departure from the ghost network, a persistent feature of existing methods. We are confused by the statement of the reviewer that "the gradient for a particular weight is not defined by its current value" - this seems to ignore the fact that normally weights influence their gradients indirectly by influencing the forward pass, which is not the case for the magnitude of latent weights. Furthermore, although Bop shares the use an exponential moving average (EMA) of gradients with Momentum, here it is motivated by the need for signal consistency (line 135-139), rather than the curvature of a smooth loss landscape, as such a landscape does not exist here. Finally, it is true that the EMA in Bop plays an analogous role to the latent weights in existing methods. However, in previous methods, it is standard to use Momentum or Adam *on top of* latent weights, creating a stacking of effects that is difficult to reason about (as well as creating higher memory requirements during training). **Finetuning Viewpoint.** Although the reviewer agrees with our reinterpretation of the latent variables, he or she later states to perceive the problem as a finetuning issue and critiques our argument in 87-92 as misleading. This argument conclusively demonstrates a closer approximation to the latent weights is not necessarily better. The implications of this observation can be debated. However, we think the observation is relevant in this context and fits very well with the results of Merolla et al. (2016, line 85-86) and the implications of Theorem 1.

**Response to Reviewer #4. Understanding Latent Weight Methods.** It is interesting that using Momentum and Adam in latent-weight methods appears so important, even though we would argue it is essentially applying "momentum to inertia". Some elaboration on line 96-98 may be illuminating. As long as the sign of the latent weight does not change, spreading out gradients over time as in Momentum does not change the behavior of the network. Therefore, the value of using momentum must lie in the behavior of a latent weight that crosses zero. As the forward pass changes, the gradient may reverse. When using plain SGD, there is a risk of ill-behaved weights that rapidly jump back and forth, generating noisy behavior that harms training. We believe Momentum mitigates this behavior. Similarly, the threshold in Bop prevents rapid flipping of weights even if the gradient reverses. [See also response to Reviewer #3]

**Note to Area Chairs.** We have presented a novel perspective on BNN optimization and a novel optimizer. Reviewer #3 in particular seems to doubt the value of the presented ideas. Fundamentally, the question at stake here is whether the perspective of a finetuned latent network will prove limiting as a guiding principle for the development of training methods for BNNs. Any answer at this point can only be speculative. Nevertheless we hope the improved empirical results and additional clarifications will convince the reviewers the presented ideas form a promising alternative perspective. We only investigated two architectures, but Bop does show a minor improvement upon existing results in both cases. As we discussed in the original submission, we see Bop as first step along a promising path, and believe the competitive results that have been achieved are very encouraging.

[Meta-Review · NeurIPS 2019]

This paper proposed a new training method for neural networks with binary weights. The main idea is to not use the existing "latent weights approach" which treats the weights as continuous, rather a new method that relies on the sign of the weights. The proposed approach is based on momentum. Before rebuttal, the authors found the paper to be original, novel, and also simpler than existing methods. They had some concerns regarding the experiments and also a few other small concerns. Post rebuttal, their views changed and they have increased their scores. One main reason is due to the ImageNet result. However, some other concerns remains and authors are encouraged to resolve those in the final version of the paper if accepted. In its current state, I think the paper can be a good addition to NeurIPS. Therefore, I vote for an acceptance.